# Sex-Dependent Altered Expression of Cannabinoid Signaling in Hippocampal Astrocytes of the Triple Transgenic Mouse Model of Alzheimer’s Disease: Implications for Controlling Astroglial Activity

**DOI:** 10.3390/ijms241612598

**Published:** 2023-08-09

**Authors:** Beatriz Pacheco-Sánchez, Rubén Tovar, Meriem Ben Rabaa, Lourdes Sánchez-Salido, Antonio Vargas, Juan Suárez, Fernando Rodríguez de Fonseca, Patricia Rivera

**Affiliations:** 1Unidad de Gestión Clínica de Salud Mental, Instituto de Investigación Biomédica de Málaga y Plataforma en Nanomedicina-IBIMA Plataforma BIONAND, Hospital Universitario Regional de Málaga, 29010 Málaga, Spain; bea_pasan@hotmail.com (B.P.-S.); rubentovar7@hotmail.com (R.T.); meriem.benrabaa@outlook.com (M.B.R.); lourdesmedia25@hotmail.com (L.S.-S.); antoniovargasfuentes@gmail.com (A.V.); juan.suarez@ibima.eu (J.S.); 2Molecular Biotechnology, FH Campus Wien, University for Applied Sciences, Favoritenstraße 222, 1100 Vienna, Austria; 3Departamento de Anatomía Humana, Medicina Legal e Historia de la Ciencia, Universidad de Málaga, 29010 Málaga, Spain

**Keywords:** 3×Tg-AD-AD mice, Alzheimer’s disease, astrocytes, hippocampus, endocannabinoid system

## Abstract

Alzheimer’s disease (AD) is a common neurodegenerative disease. In AD-associated neuroinflammation, astrocytes play a key role, finding glial activation both in patients and in animal models. The endocannabinoid system (ECS) is a neurolipid signaling system with anti-inflammatory and neuroprotective properties implicated in AD. Astrocytes respond to external cannabinoid signals and also have their own cannabinoid signaling. Our main objective is to describe the cannabinoid signaling machinery present in hippocampal astrocytes from 3×Tg-AD mice to determine if they are actively involved in the neurodegenerative process. Primary cultures of astrocytes from the hippocampus of 3×Tg-AD and non-Tg offspring were carried out. We analyzed the gene expression of astrogliosis markers, the main components of the ECS and Ca^2+^ signaling. 3×Tg-AD hippocampal astrocytes show low inflammatory activity (*Il1b*, *Il6*, and *Gls*) and Ca^2+^ flow (*P2rx5* and *Mcu*), associated with low cannabinoid signaling (*Cnr1* and *Cnr2*). These results were more evident in females. Our study corroborates glial involvement in AD pathology, in which cannabinoid signaling plays an important role. 3×Tg-AD mice born with hippocampal astrocytes with differential gene expression of the ECS associated with an innate attenuation of their activity. In addition, we show that there are sex differences from birth in this AD animal, which should be considered when investigating the pathogenesis of the disease.

## 1. Introduction

Alzheimer’s disease (AD) is a progressive disorder that is characterized clinically by memory loss and pathologically by extracellular accumulation of beta-amyloid peptide (Aβ), neurofibrillary tangle formation, extensive neuroinflammation, synaptic dysfunction, neurodegeneration, and brain dysfunction [1,2,3]. The number of patients with Alzheimer’s disease (AD) shows a very rapid growth trend, resulting in a deterioration in the quality of daily life and a very high economic cost for public health. The greatest risk factor for AD is aging, although there are others, such as an indirect family history of AD, heredity, and lifestyle; however, the etiology of AD is not fully understood.

Neuroinflammation, in which astrocytes and microglia play a key role, has recently been associated with AD pathogenesis. It was shown that this is not simply a consequence of the deposition of beta-amyloid plaques and neurofibrillary tangles [4,5].Thus there is glial activation in both patients and animal models of AD [6,7,8,9,10].

Astrocytes, the most abundant type of glial cells, are specialized cells with a structural function that maintain brain homeostasis by providing trophic and metabolic support to neurons, playing a critical role in synaptic transmission and plasticity. This astrocytes–neurons interaction is essentially controlled by intracellular Ca^2+^ concentrations [11]. Astroglial cells are key regulators of neuroinflammation, since they can release and respond to a wide variety of cytokines. Astrocytes are found around senile plaques phagocytizing toxic compounds such as Aβ; however, they also produce cytotoxic factors and promote the neurodegeneration observed in AD [2,12,13].

One of the key regulatory systems of astrocyte physiology is the endocannabinoid system (ECS) [14,15]. ECS is a neurolipidic signaling system that acts as a modulator of physiological homeostasis. Its alteration has been implicated in various pathophysiological events of the central nervous system (CNS) and periphery including obesity, neurodegeneration, inflammatory diseases, and psychiatric disorders [16,17,18,19]. Numerous studies have shown the involvement of the ECS in Alzheimer’s disease, proposing it as a target for treatment given its anti-inflammatory and neuroprotective properties [20,21,22].

This signaling system is composed of the endogenous lipidic substances with cannabimimetic effects called endocannabinoids [23], anandamide (N-arachidonoyl-ethanolamine, AEA), and 2-arachidonoyl-glycerol (2-AG), which are best-characterized transmitters interacting with cannabinoid CB1 and CB2 receptors. Endocannabinoid signaling also acts through other non-cannabinoid receptors, including nuclear receptors (PPARα and γ receptors) and transient receptor potential channel ionotropic receptors (TPRV1) [24,25]. Endocannabinoids are synthesized on demand by enzymes responsible for the synthesis of the two main endocannabinoids, N-arachidonoylphospatidylethanolamine phospholipase D (NAPE-PLD) and diacylglycerol lipase (DAGLα and DAGLβ). The degradative enzymes, fatty acid amide hydrolase (FAAH), and monoacylglycerol lipase (MAGL) play an important role in regulating endocannabinoid activity [26,27]. AEA and 2-AG share metabolic pathways with other endocannabinoid-like lipid mediators, such as palmitoylethanolamide (PEA) and oleoylethanolamide (OEA), which act on their own non-cannabinoid target molecules (i.e., PPARα).

Astrocytes not only respond to cannabinoid signals, but also synthesize and release endocannabinoids [11]. Endocannabinoids modulate astrocyte physiology due to interaction with cannabinoid receptors (CB1, GPR55, and maybe CB2) present in them; this modulation is carried out through several mechanisms, including the direct control of astroglial glucose metabolism by CB1 receptors located in mitochondrial membranes [14,15] or the mobilization of Ca^2+^ from the astrocytes by the CB1 receptor [28].

Numerous studies have shown the great potential of ECS to control degenerative diseases such as AD, due to its anti-inflammatory and neuroprotective properties that involve astrocyte engagements. Interestingly, cannabinoid signaling in glial cells essential has this function, since neuroimmunomodulation of glial cells by exogenous cannabinoid treatment enhances behavioral and cognitive responses, resulting in improved learning and memory-related symptoms in AD (for review see [29]).

The present study describes the cannabinoid signaling machinery present in cultured hippocampal astrocytes obtained from 3×Tg-AD mice. This so-called “triple transgenic mouse model” mimics many critical aspects of AD neuropathology. They were engineered to express a transgene harboring the APPSwe, PS1M146V, and tauP301L mutations of the amyloid precursor protein (APP), presenilin-1, and microtubule-associated protein tau, respectively [30]. By culturing astrocytes from these animals, which express the transgene, we can provide insight into whether astrocytes are innocent bystanders that react to the damage that occurs during AD or are actively involved in the neurodegenerative process. In addition, we can analyze the differential expression of cannabinoid signaling machinery by comparing control versus 3×Tg-AD mice astrocytes.

There are studies that show that the 3×Tg-AD animal model presents cerebral alterations from early stages of life, even before the appearance of the pathologies typical of the development of the disease. Thus, 3×Tg animals present neuroanatomical alterations before the manifestation of Alzheimer’s behaviors, suggesting that the mutations in this animal model are sufficient to cause changes in neuroanatomy in 3×Tg mice, but potentially insufficient to be responsible for behavioral changes in the first years of life [31].

3×Tg-AD animals have also been shown to exhibit neurogenesis defects from early postnatal stages, long before the onset of any neuropathology or behavioral deficits. Furthermore, these animals have fewer neural stem/progenitor cells, with less proliferation and reduced numbers of newborn neurons in the postnatal stages, consistent with reduced hippocampal volumes [32].

Regarding the glia, other authors have described that in 3×Tg-AD animals, there is an astroglial alteration from an early age. Thus, these animals present a morphological alteration of astrocytes, associated with a functional deficit, from before the appearance of amyloid beta deposits and senile plaques [33,34].

There are studies that use primary cultures of astrocytes from the 3×Tg-AD animal model, most of them exposing them to Aβ42, with the aim of testing the neuroprotective capacity of cannabinoid compounds [6,20]. However, there are no studies that characterize primary astrocytes per se of this animal model in terms of the phenotypic-related cannabinoid signaling resulting from the expression of human transgenes for AD.

The importance of glial implication in AD pathology has been increasingly recognized, but ECS–glial interaction occurring in AD requires further study.

Our results demonstrate for the first time that 3×Tg-AD mice are born with attenuated hippocampal astrocytes in terms of their inflammatory activity and Ca^2+^ signaling-related proteins, which seem to be associated with low endocannabinoid signaling. In addition, there is a clear sexual dimorphism, since these findings were more evident in females.

## 2. Results

### 2.1. mRNA Expression of Inflammation Markers

To analyze whether hippocampal astrocytes from 3×Tg-AD mice present greater inflammatory reactivity or astrogliosis, we analyzed gene expression of inflammatory markers and markers indicative of glial reactivity: glial fibrillary acidic protein (*Gfap*), vimentin, tumor necrosis factor α (*Tnfα*), interleukins (*Il1b* and *Il6*), and prostaglandin-endoperoxide synthase 2 (Ptgs or Cox2).

Two-way ANOVA indicated a sex effect on *Tnfa* mRNA levels [F (1,28) = 6.865; *p* < 0.05], with decreased *Tnfa* mRNA levels in hippocampal astrocytes of 3×Tg-AD female offspring compared to 3×Tg-AD male astrocytes (uncorrected Fisher’s LSD post hoc test; * *p* < 0.05; Figure 1C).

An interaction between sex and genotype [F (1,28) > 4.677; *p* < 0.05] was found on *Il1b* mRNA levels. No differences were found between groups using Tukey’s multiple comparisons test (Figure 1D).

A genotype effect on *Il6* mRNA expression was observed [F (1,28) > 7.703; *p* < 0.01], with an overall decrease in *Il6* expression in 3×Tg-AD female astrocytes compared to non-Tg (WT) female and 3×Tg-AD male groups (uncorrected Fisher’s LSD post hoc test; *^/##^
*p* < 0.05/0.01; Figure 1E).

No effects of the factors analyzed on the expression of *Gfap, Vimentin*, and *Ptgs* were found.

### 2.2. mRNA Expression of Cannabinoid-Related Receptors

Two-way ANOVA indicated an interaction between sex and genotype [F (1,28) = 13.65 3; *p* < 0.001] on *Cnr1* mRNA levels, with WT female astrocytes showing higher *Cnr1* levels than WT male astrocytes (* *p* < 0.05; Figure 2A). Tukey’s multiple comparisons test also showed decreased *Cnr1* expression in hippocampal astrocytes from female offspring born to 3×Tg-AD mothers compared to WT female astrocytes (^#^
*p* < 0.05; Figure 2A).

A genotype effect on *Cnr2* mRNA expression was found [F (1,28) = 10.31; *p* < 0.01]. An interaction between the two factors analyzed was also found in the *Cnr2* expression [F (1,28) = 4.213; *p* < 0.05]. Hippocampal astrocytes from 3×Tg-AD female offspring had a lower level of *Cnr2* than WT female astrocytes (Tukey’s multiple comparisons test; ^##^
*p* < 0.01; Figure 2B). We also observed an increased in *Cnr2* expression in female WT (non-Tg) astrocytes compared to the WT male group (* *p* < 0.05; Figure 2B).

No effects of the factors analyzed on the expression of *Gpr55*, *Ppara*, *Trpv1*, and *Trpa1* were found.

### 2.3. Cannabinoid Enzymes mRNA Expression

Regarding the enzymatic machinery for endocannabinoid biosynthesis and degradation, an interaction between sex and genotype was found on *Daglb* mRNA levels [F (1,28) = 6.881; *p* < 0.05], with an overall decrease in *Daglb* mRNA levels in 3×Tg-AD female astrocytes compared to 3×Tg-AD male group (Tukey’s multiple comparisons test; * *p* < 0.05; Figure 3B).

Two-way ANOVA also showed an interaction between two factors analyzed on *Nape-pld* mRNA levels [F (1,28) = 4.167; *p* < 0.05].

No effects of the factors analyzed on the expression of *Dagla*, *Mgll*, and *Faah* were found.

### 2.4. Cannabinoid Enzymes Protein Expression

Two-way ANOVA showed a genotype effect [F (1,20) = 5.558; *p* < 0.05] and an interaction between sex and genotype in MAGL protein expression [F (1,20) = 4.283; *p* < 0.05]. Tukey post hoc analysis indicated a decreased of MAGL expression in hippocampal astrocytes from female offspring born to 3×Tg-AD mothers compared to WT female astrocytes (^##^
*p* < 0.01; Figure 4B).

A genotype effect on FAAH protein expression was found [F (1,20) = 6.416; *p* < 0.05]. The uncorrected Fisher’s LSD post hoc test showed lower FAAH protein level in 3×Tg-AD male astrocytes compared to WT male group (^##^
*p* < 0.01; Figure 4E).

To analyze the balance between endocannabinoid production and degradation, we calculated the ratio between the enzymes of synthesis and degradation of acylethanolamides and acylglycerols.

Two-way ANOVA showed an interaction between sex and genotype in DAGLα/MAGL ratio [F (1,20) = 7.081; *p* < 0.05]. Tukey post hoc analysis indicated an increased DAGLα/MAGL ratio in 3×Tg-AD female astrocytes compared to 3×Tg-AD male and WT female groups (*^/#^
*p* < 0.05; Figure 4F).

We also found a genotype effect [F (1,20) = 7.187; *p* < 0.05] and an interaction between sex and genotype [F (1,20) = 4.125; *p* < 0.05] in the NAPE-PLD/FAAH ratio. Tukey post hoc analysis indicated that hippocampal astrocytes from 3×Tg-AD male offspring had a higher NAPE-PLD/FAAH ratio than WT male astrocytes (^#^
*p* < 0.05; Figure 4H).

No effects of the factors analyzed on the expression of DAGLα, MAGL, and DAGLβ/MAGL were found.

### 2.5. Culture Medium Endocannabinoids Levels

Regarding the endocannabinoid levels measured in the culture medium: 2-Arachidonoylglycerol (2AG), 2-Linoleoylglycerol (2LG), palmitoleoylethanolamide (POEA), linoylethanolamide (LEA), palmitoylethanolamide (PEA), oleoylethanolamide (OEA), and N-stearoylethanolamine (SEA).

A main effect of genotype on LEA levels was found [F (1,16) = 4.638; *p* < 0.05]. Although the uncorrected Fisher’s LSD test did not show significant differences between groups, we observed an overall increase in LEA levels in 3×Tg-AD astrocytes compared to WT (non-Tg) (male 3×Tg-AD vs. male WT *p* = 0.1280; female 3×Tg-AD vs. female WT *p* = 0.1690; Figure 5D).

Two-way ANOVA also showed a genotype effect on POEA and OEA levels [F (1,16) > 4.963; *p* < 0.05], with an overall decrease in POEA and OEA levels in 3×Tg-AD astrocytes compared to WT, although these differences are not significant applying the uncorrected Fisher’s LSD post hoc test (POEA: male 3×Tg-AD vs. male WT *p* = 0.1498; female 3×Tg-AD vs. female WT *p* = 0.1428; Figure 5C) (OEA: male 3×Tg-AD vs. male WT *p* = 0.1245; female 3×Tg-AD vs. female WT *p* = 0.1521; Figure 5F).

No effects of the factors analyzed on 2AG, 2LG, PEA, and SEA levels were found. Despite the non-statistical significance of the results, we also observed evident differences between the groups analyzed in the levels of 2AG (male 3×Tg-AD vs. male WT *p* = 0.3255; female 3×Tg-AD vs. female WT *p* = 0.4667; Figure 5A)

### 2.6. mRNA Expression of Ca^2+^ Signaling-Related Genes

A genotype effect on the mRNA levels of the ionotropic channel *P2x5* [F (1,28) = 5.152; *p* < 0.05] and an interaction between sex and genotype [F (1,28) = 18.11; *p* < 0.001] were also found. Post hoc analysis (Tukey’s multiple comparisons test) showed higher levels of *P2x5* in WT female astrocytes than males but decreased *P2x5* mRNA expression in 3×Tg-AD female astrocytes than 3×Tg-AD male group (* *p* < 0.05; Figure 6A). We also found that hippocampal astrocytes from 3×Tg-AD female offspring had a lower level of *P2x5* than WT female astrocytes (^###^
*p* < 0.001; Figure 6A).

We also found interaction between sex and genotype [F (1,28) = 14.18; *p* < 0.001] in the mRNA levels of the mitochondria channel calcium *Mcu*, with an overall decrease in *Mcu* mRNA levels in hippocampal astrocytes from 3×Tg-AD female offspring (Tukey’s multiple comparisons test; *^/#^
*p* < 0.05; Figure 6B).

To study a possible glutamate toxicity, we have analyzed the expression of two isoforms of glutaminase, which generates glutamate from glutamine. We observed an interaction between the two factors analyzed in *Gls* mRNA expression [F (1,28) = 21.23; *p* < 0.001], with 3×Tg-AD female astrocytes showing a lower level of *Gls* than WT female and 3×Tg-AD male groups (Tukey’s multiple comparisons test; **^/###^
*p* < 0.01/0.001; Figure 6E).

No effects of the factors analyzed on the expression of *Nsmf*, *Itpr1*, and *Gls2* were found.

## 3. Discussion

It is increasingly evident that glial cells are implicated in the pathology of AD. It has also been described that the ECS, due to its neuroprotective function, plays a modulatory role in the development and expression of the disease [35]. However, although astrocytes respond to cannabinoid signals and have their own intrinsic signaling, the ECS–glial interaction that occurs in AD is poorly studied. Here, we describe cannabinoid signaling in hippocampal astrocytes of 3×Tg-AD mice, the triple transgenic mouse model, demonstrating for the first time that female 3×Tg-AD mice present at birth (PND 2-3) in the hippocampus astrocytes with a lower expression of both genes related to inflammatory activity (*Il1b* and *Il6*; Figure 1) and genes related to Ca^2+^ flow (*P2rx5* and *Mcu*; Figure 6) located at the plasma membrane and subcellular levels. This astroglial attenuation is associated with the low expression of several cannabinoid signaling-related genes, highlighting the decrease in gene expression of the CB1 and CB2 receptors (Figure 2). In addition, these effects were sex-dimorphic, since these results were more evident in females.

Astrocytes modulate neuroinflammation, not only by responding to inflammatory mediators, but also by producing cytokines and chemokines involved in both protective and toxic roles in neuroinflammatory processes [36]. In pathological conditions, such as AD, astrocytes undergo reactive astrogliosis, identified by overexpression of glial fibrillary acidic protein (GFAP) and vimentin in post-mortem tissues from AD patients and mouse models [37,38]. This reactivity is also associated with an increased production of proinflammatory mediators [13]. Our results show that hippocampal astrocytes from 3×Tg-AD offspring at postnatal day 2–3 do not present overexpression of GFAP or Vimentin (Figure 1). Thus, AD-associated mutations do not induce reactive astrogliosis in vitro at birth, so this phenomenon will probably appear throughout the course of the disease. Most mouse models for AD show reactive astrogliosis commonly associated with amyloid plaques and hyperphosphorylation and oligomerization of tau. This astrocyte reactivity increases with age in the hippocampus, amygdala, and entorhinal cortex [39,40,41,42].

Oxidative stress and inflammation are closely implicated in the progression of various neurodegenerative diseases, including AD. The Aβ1-42 peptide induces the expression of markers of inflammation and oxidative stress in primary cultures of astrocytes such as IFNγ, IL-1β, TNFa, IL-6, and TGFβ, being also overexpressed in human brain samples with AD and in animal models [20,37]. Interestingly, we have found that hippocampal astrocytes from 3×Tg-AD mice at 2–3 postnatal days show a decrease in the mRNA expression of inflammation-related markers, such as *Il1b* and *Il6* (Figure 1D,E). The role of reactive astrogliosis in CNS diseases is complex. Reactive astrocytes are needed to repair damage, for example, surrounding Ab plaques in AD; however, reactive astrocytes can be neurotoxic by producing reactive oxygen species and inflammatory cytokines [37]. Our results lead us to hypothesize an innate attenuation of astrocytic activity in this animal model of AD. We can hypothesize that this situation might eventually impact the defensive reactivity against the damage produced along the development of the disease, although this hypothesis needs to be confirmed experimentally. However, this observation contributes to gaining insight into understanding the functional consequence of reactive astrocytes in AD.

Interestingly, these changes are mostly sex-dependent; the females are more affected by the genotype factor, setting in place the question of the sex dependency on the astrocyte role in AD. Many neurological disorders exhibit sex differences and sex-specific therapeutic responses, including Alzheimer’s disease. However, most of the studies that use cell cultures to study the mechanisms underlying the disease do not take into account the sex of the animal of origin, which can lead to contradictory and inconclusive results. Although mammalian sex differences become apparent around adolescence, many studies show that the brain of neonatal mice is different in males and females in terms of molecular and cellular mechanisms and in response to perinatal treatment/insults [43,44,45]. Recently, Zhang et al. [46] published a study that identifies sexually dimorphic genes in primary cultures of neurons, astrocytes, and microglia, demonstrating that sex should be taken into account in primary cultures when studying neurological disorders.

In recent decades, extensive research has been carried out on the role of endocannabinoid signaling in brain pathology, given its neuroprotective role, with therapeutic objectives. ECS alterations in neurological disorders include changes in CB1 and/or CB2 receptor expression and endocannabinoid levels, but their impact the onset and progression of the disease remains unclear due to the complexity of the endocannabinoid signaling and the pathophysiology of neurodegenerative diseases.

Endocannabinoid signaling in astrocytes is demonstrated by in vitro and in vivo studies, as well as in vivo analysis of the effects of endocannabinoid-modulating drugs under physiological and disease conditions [6,20,28].

Several studies show a deregulation of the expression of CB1 in Alzheimer’s disease; however, while in animal models it seems clear that there is a decrease in this receptor [35,47,48], in humans, there is more controversy, since its expression depends on the stage of the disease [22]. Our results confirm a decrease in CB1 gene expression (Figure 2A) in the hippocampal astrocytes from 3×Tg-AD mice at 2-3 PND; moreover, this decrease is specific to females. Interestingly, Kalifa et al. [48] associated reduced CB1 expression in an AD mouse model with increased astrocytes and their reactivity; however, the reduction in CB1 receptors did not appear to be astrocyte-specific but neuronal in this study. In this work, we demonstrate that there is a decrease in the gene expression of CB1 specific to astrocytes in the animal model of Alzheimer’s used (Figure 2A).

Similarly, we found a reduction in CB2 gene expression (Figure 2B) that is also sex-dependent (more evident in females). However, we have not found alterations in the gene expression of the other receptors related to cannabinoid signaling (Gpr55, Ppara, Trpv1, and Trpa1; Figure 2C–F), which indicates that astrocytic activity in this Alzheimer’s animal model is directly controlled via CB1 and CB2. To date, the relationship of these receptors with the pathogenesis of Alzheimer’s has hardly been studied. TRPV1 was analyzed in human samples without finding differences between Alzheimer’s patients and healthy individuals [22]. It is hypothesized that the glial CB2 receptor regulates astrogliosis in the presence of plaques and tangle pathology in AD. Thus, studies in rodents and humans show a high astroglial CB2 expression correlated with the amyloid deposit [49]. This hypothesis is consistent with the low CB2 expression found in our study in 3×Tg-AD mouse astrocytes at birth, since these are not activated, as we have pointed out. Therefore, CB2 could be used as a therapeutic target to restrict inflammatory processes that contribute to AD progression. In this regard, the use of synthetic cannabinoids, such as the mixed CB1/CB2 agonistWin55, 212-2, or the CB2 selective agonist JWH-133, has been shown to improve memory and decrease neuroinflammation in vivo and in vitro models of AD [20,50,51].

Cannabinoid enzymes and ligands also appear to be involved in the development of AD. Benito et al. [41] showed that FAAH (expression and activity) is abundantly and selectively expressed in neuritic plaque-associated astrocytes in post-mortem brains of patients with AD. N-acylethanolamines are substrates of FAAH, which release the fatty acid moiety. A similar effect is produced by the action of MAGL on acylglycerols. These fatty acids released subsequently can be converted in additional inflammation-triggering compounds, such as the case for prostaglandins derived from arachidonic acid. In our study, we did observe minor sex-dependent dimorphism on FAAH protein expression in astrocytes that is dependent on the AD phenotype. Interestingly, the ratio production/degradation of ECS (measured as the DAGL/MAGL and NAPE-PLD/FAAH ratios), is affected differentially in males and females, although its significance cannot be deducted from the present data. What is remarkable is that these alterations are associated with changes in the concentration of specific acylethanolamides measured. While the proinflammatory LEA (which is also a competitive inhibitor of FAA) is clearly enhanced, the anti-inflammatory OEA and POEA are markedly reduced, suggesting a dysregulation of the modulatory role of these lipids on inflammation. The mechanistic of the differential release of acylethanolamides species is currently unknown, demanding more active research.

Astrocytes communicate with the rest of the cells through mechanisms regulated mainly by intracellular Ca^2+^; which can be controlled in turn by cannabinoid signaling [11,28]. For this reason, we have also analyzed the mRNA expression of some important players in astrocytic Ca^2+^ dynamics (Figure 6).

We found a profile similar to that of cannabinoid receptors, with an overall decrease in the main Ca^2+^ channels at both plasma level (P2rx5 purinergic receptor and NMDA receptor) and subcellular level (Mitochondrial Ca^2+^ uniporter, Mcu; and Inositol triphosphate receptor type 1 in the endoplasmic reticulum, Itpr1) in the hippocampal astrocytes of 3×Tg-AD female animals at PND 2-3.

It has been shown that activation of CB1 increases the Ca^2+^ levels of astrocytes. In addition, CB1 receptors in astrocytes are also found in the mitochondrial membrane (mtCB1) [15], and its activation enhances Ca^2+^ flux between the endoplasmic reticulum and mitochondria; moreover, MCU inhibition affects subcellular spread of Ca^2+^ [11]. All these findings suggest that 3×Tg-AD female astrocytes also present an attenuation of the subcellular Ca^2+^ transit regulated by mtCB1, presenting a low profile of all the calcium receptors (*Mcu* and *Itpr1*; Figure 6B,D) involved as well as of CB1.

Glutamate is the main excitatory neurotransmitter; however, it can induce neuronal damage through excitotoxicity, which results from overactivation of glutamatergic receptors. This glutamate toxicity is implicated in the pathogenesis of AD and astrocytes may play a key role in it. Glutamate release from astrocytes contributes to NMDA-related long-term depression. This mechanism is controlled by CB1 activation of astrocytes, increasing the concentration of Ca^2+^, and thus, producing the release of astrocytic glutamate [52]. The hippocampal astrocytes of the AD animals present low expression of glutaminase (*Gls*; Figure 6E) as well as of the NMDA receptor (*Nsmf*; Figure 6C), especially in females, which could be indicating a low release of glutamate by these astrocytes. Therefore, in this sense, there is no astrocytic involvement in the development of AD mediated by glutamate excitotoxicity.

Finally, it should be noted that most of the results presented show a clear sexual dimorphism, with changes in the markers analyzed being more present in females than in males. We found an interaction of sex and genotype in many of our results, i.e., the changes induced by genotype are sex-dependent.

In humans, a sex difference in AD prevalence is described, but less so in brain pathology and underlying molecular mechanisms. In the 3×Tg-AD animal model, female mice show more amyloid plaques, neurofibrillary tangles, neuroinflammation, and cognitive deficits than males at 12 months of age [53]. Our study now provides that sex differences in this animal model of AD are evident from birth and, therefore, sex differences should be taken into account when investigating the pathogenesis of the disease.

The present study has several limitations. First, we were unable of measuring anandamide because of technical reasons. Anandamide is a relevant modulator of astrocytic networks through its role in the modulation of gap junctions-mediated calcium signals [54]. Second, this study is not a dynamic analysis of astrocytic response. Thus, future studies should introduce the time course of astrocyte activation. Third, we did not examine the status of mitochondrial CB1 receptor-regulated genes that, indeed, may play an important role in differential astrocyte physiology in control versus 3×Tg-AD mice. Fourth, another limitation of the present study is its potential dependence on the genetic model used. Future studies using differential models (i.e., accelerated amyloidosis models, such as 5xFAD, or APP knock-in models), should address the existence of astrocytic alterations early in the development as a hallmark of AD.

An important aspect not studied in the present work is the role of the alternative pathway for enocannabinoid synthesis and metabolism, involving the ABHD family of enzymes. The availability of free arachidonic or other poly-unsaturated fatty acids potentially engaging with the formation of prostaglandins and endoperoxides is very relevant and deserves further research.

In any case, the data are sufficient to support the existence of sex and genotype dimorphic alterations in astrocytes, demanding a carefully differential experimental approach.

## 4. Materials and Methods

### 4.1. Ethics Statement

All procedures were conducted in strict adherence to the principles of laboratory animal care (National Research Council, Neuroscience CoGftUoAi, Research B, 2003) following the European Community Council Directive (86/609/EEC) and were approved by the Ethical Committee of the University of Málaga. Special care was taken to minimize the suffering and the number of animals needed to perform the procedures.

### 4.2. Animal Model

The 3×Tg-AD-AD (Jackson Laboratory) mouse model combines mutant hAPP (Swedish), PSEN1 (MM146V), and tau (P301L) transgenes, resulting in Aβ and tau pathologies.

Non-transgenic (wild type or non-Tg; WT) and transgenic (3×Tg-AD) female adolescent mice were individually housed in standard cages and maintained under controlled room conditions: 21 ± 1 °C room temperature, 40 ± 5% relative humidity, and a 12-h light-dark cycle (lights off at 8:00 p.m.).

Females were allowed to mate with males of the same strain in their home cage for 72 h. The presence of a spermatozoa plug in the vaginal smear confirmed successful mating, which was designated as gestational day 0 (GD0). At postnatal day 2–3, 3×Tg-AD and WT offspring (male and female) were sacrificed by decapitation.

### 4.3. Primary Cultures of Hippocampal Astrocytes

Primary astrocytes were derived from male and female pups at PND2 separately (WT male: *n* = 8; 3×Tg-AD male: *n* = 8; WT female: *n* = 8; 3×Tg-AD female: *n* = 8), as described previously [55,56]. Briefly, primary cultures were generated from the hippocampus and maintained in DMEM:F12 (Gibco, Grand Island, NY, USA) with 10% fetal bovine serum (FBS) and 1% antibiotic antimycotic solution. After 10 days in vitro, when astrocyte cultures were 70–80% confluent, other cell types were removed by orbital rotation for at least 16 h at 280 rpm at 37 °C. This procedure results in enrichment of astrocytes to a purity of at least 95%. The remaining cells were plated at a density of 4.35 × 105 cells/cm^2^ and cultured for 24 h. They were then grown in serum free media for 24 h before freezing.

### 4.4. RNA Isolation and Real-Time Quantitative PCR Analysis

Real-time PCR was performed using specific sets of primer probes from TaqMan Gene Expression Assays (ThermoFisher Scientific, Waltham, MA, USA; Table 1). RNA from astrocytes was extracted following the Trizol method according to the manufacturer’s instructions (ThermoFisher Scientific). A melting curve analysis was performed to ensure that only a single product was amplified. After analyzing several control genes, values obtained from the samples were normalized to beta-actin *(Actb*) levels, which did not vary significantly between groups.

### 4.5. Western Blot Analysis

Western blotting was performed as described previously [57]. Briefly, astrocyte cultures (*n* = 6) were homogenized in 500 µL of ice-cold lysis buffer containing Triton X-100, 1 M 4-(2-hydroxyethyl)-1-piperazineethanesulfonic acid (HEPES), 0.1 M ethylenediaminetetraacetic acid (EDTA), sodium pyrophosphate, sodium fluoride (NaF), sodium orthovanadate (NaOV), and protease inhibitors using a tissue-lyser system (Qiagen, Hilden, Germany). After centrifuging at 26,000× *g* for 30 min at 4 °C, the supernatant was transferred to a new tube. The Bradford method was used to measure the protein concentration of the samples. A quantity of 30 µg of each total protein sample was separated on 4–12% polyacrylamide gradient gels. The gels were then transferred onto nitrocellulose membranes (Bio-Rad Laboratories, Hercules, CA, USA) and stained with Ponceau red, Membranes were blocked in TBS-T (50 mM Tris-HCl pH 7.6, 200 mM NaCl, and 0.1% Tween 20) with 2% albumin fraction V from bovine serum (BSA, Roche, Mannheim, Germany) for 1 h at room temperature. The primary antibodies to the proteins of interest (Table 2) were incubated overnight at 4 °C. Mouse βactin was used as the reference protein. After several washes in TBS containing 1% Tween 20, an HRP-conjugated anti-rabbit or anti-mouse IgG (H + L) secondary antibody (Promega, Madison, WI, USA), diluted 1:10,000, was added followed by incubation for 1 h at room temperature. After extensive washing in TBS-T, the membranes were incubated for 1 min with the Western Blotting Luminol Reagent kit (Santa Cruz Biotechnology, Santa Cruz, CA, USA), and the specific protein bands were visualized and quantified by chemiluminescence using a ChemiDocTM MP Imaging System (Bio-Rad, Barcelona, Spain). The results are expressed as the target protein/βactin ratios.

### 4.6. Endocannabinoid Quantification

Culture medium samples were analyzed following the method indicated by Pastor et al. [58]. For the preparation of the samples, 500 μL of volume were used and two replicates were performed, which was measured three times each in an HPLC-MS. The samples were spiked with a mix of internal standards with a concentration between 1.5 and 15 ng/L.

The chromatographic separation was carried out on a HPLC (high performance liquid chromatography tandem mass spectrometry) model ultimate 3000 from Thermo Scientific. The column was ACE Excel 2 C18 (2 μm particle size, 10 × 3.0 mm ACE) maintained at 40 °C with a mobile phase flow rate of 0.3 mL/min. The composition of the mobile phase was: A, 0.1% (*v*/*v*) formic acid in water and B, 0.1% (*v*/*v*) formic acid and 0.05 mM of NaCl in acetonitrile. The initial conditions were 60% A and 40% B. The gradient was increased linearly to 100% B over 5 min, maintained at 100% B for 3 min, and returned to the initial conditions for a further 4 min with a total run time of 20 min. The mass spectrometer was a Q Exactive from Thermo Scientific with an Orbritap detector. The spectral range was recorded between 270–410 m/z in positive mode for: DHEA, POEA, AEA, LEA, 2AG, 2LG, DEA, PEA, OEA, 2OG, and SEA. The desolvation gas temperature was 230 °C, a gas flow rate of 40 mL/min, auxiliary gas 15 mL/min, and sweep gas 2 mL/min. The spray voltage was set at 3.0 kV. The compounds were identified by their exact mass with an exactitude inferior to 2 ppm. A six-point external calibration curve was prepared with concentration between 0.5–15 ng/L for SEA, DEA, DHEA, AEA, LEA, and OEA and between 10–300 ng/L for 2AG, 2LG, 2OG, PEA, and POEA and the mix of internal standards (SEA-d4, LEA-d4, PEA-d4, OEA-d4, AEA-d4, and DHEA-d4 at a concentration of 1.5 ng/L and 2AG-d5 at a concentration of 15 ng/L, the same used in the extraction of the samples).

### 4.7. Statistical Analysis

All data are expressed as mean ± SEM. Animal model data were analyzed by two-way ANOVA (sex and genotype). IBM SPSS Statistics 23 and GraphPad Prism 6 software programs were used. Subsequent multiple comparisons between groups were carried out using Tukey adjustments or simple effect analysis (Fisher’s test) in cases of factor effect but no interaction. *p* < 0.05 was considered statistically significant.

## 5. Conclusions

In this study, we characterized the primary astrocytes of the 3×Tg-AD model in terms of cannabinoid signaling related to the phenotype resulting from the expression of human transgenes for AD. Here, we demonstrate for the first time that 3×Tg-AD mice display inflammatory activity and attenuated Ca^2+^ flux at birth in hippocampal astrocytes compared to non-transgenic animals, which appears to be associated with decreased endocannabinoid signaling. Therefore, we can ensure that astrocytes not only responded to the damage that will be generated throughout the development of the disease, but that they are actively involved in the neurodegenerative process, innately having differential activity compared to wild-type ones.

Sex differences in humans and in this animal model of AD have already been established, but always using older individuals. Most of the results presented now show a clear sexual dimorphism, with changes of the markers analyzed being more present in females; thus, the present study shows that sex differences in this animal model of AD are evident from birth and, therefore, sex differences should be taken into account when investigating the pathogenesis of the disease.

## Figures and Tables

**Figure 1 ijms-24-12598-f001:**
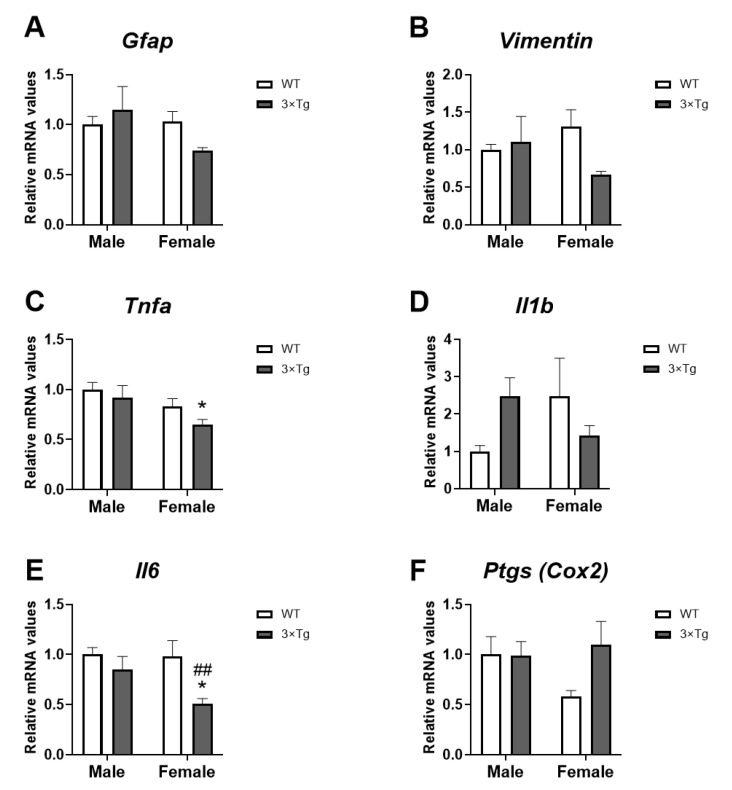
Inflammation markers’ mRNA expression: *Gfap* (**A**), *Vimentin* (**B**), *Tnfa* (**C**), *Il1b* (**D**), *Il6* (**E**), and *Ptgs* (**F**) in hippocampal astrocytes from offspring born to 3Tg-AD and non-Tg (WT) mice. Data are expressed as the mean ± S.E.M. Tukey’s or single-effect analysis: * *p* < 0.05 vs. males (sex difference); ^##^
*p* < 0.01 vs. same-sex WT group.

**Figure 2 ijms-24-12598-f002:**
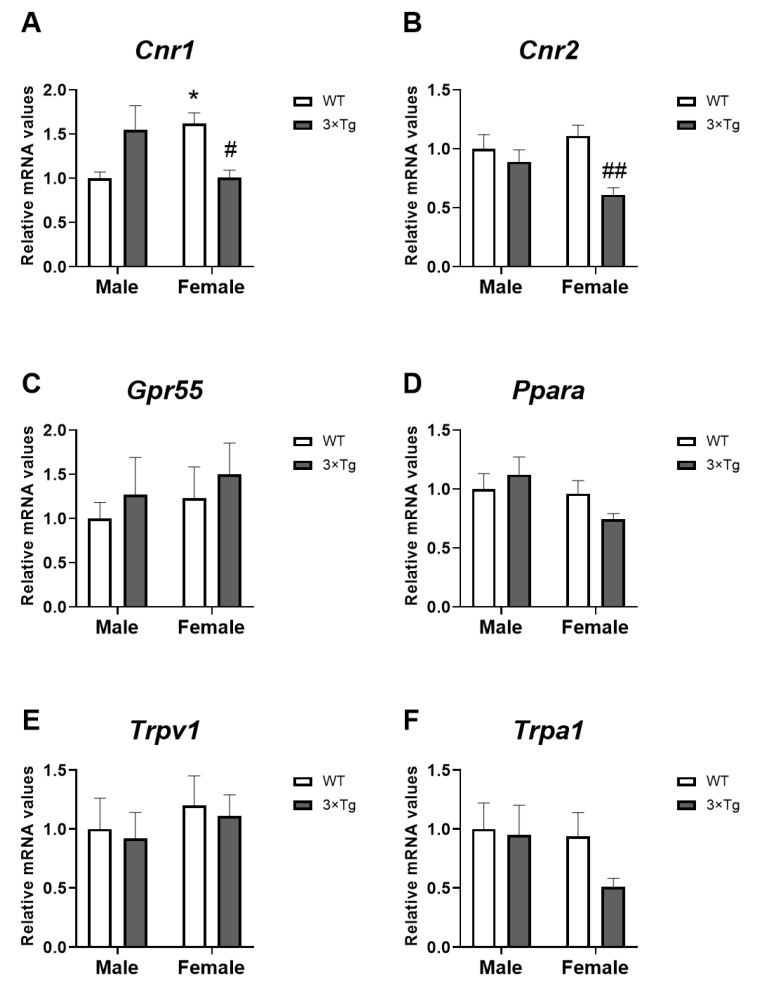
Cannabinoid-related receptors mRNA expression: *Cnr1* (**A**), *Cnr2* (**B**), *Gpr55* (**C**), *Ppara* (**D**), *Trpv1* (**E**), and *Trpa1* (**F**) in hippocampal astrocytes from offspring born to 3×Tg-AD and non-Tg (WT) mice. Data are expressed as the mean ± S.E.M. Tukey or single-effect analysis: * *p* < 0.05 vs. males (sex difference); ^#/##^
*p* < 0.05/0.01 vs. same-sex WT group.

**Figure 3 ijms-24-12598-f003:**
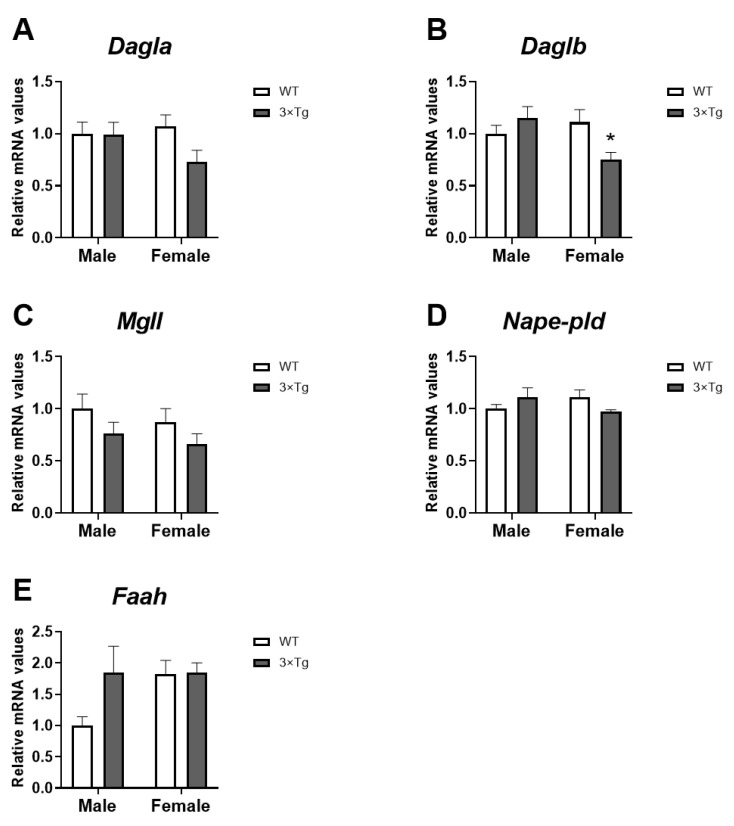
Cannabinoid enzymes mRNA expression: *Dagla* (**A**), *Daglb* (**B**), *Mgll* (**C**), *Nape-pld* (**D**), and *Faah* (**E**) in hippocampal astrocytes from offspring born to 3×Tg-AD and non-Tg (WT) mice. Data are expressed as the mean ± S.E.M. Tukey or single-effect analysis: * *p* < 0.05 vs. males (sex difference).

**Figure 4 ijms-24-12598-f004:**
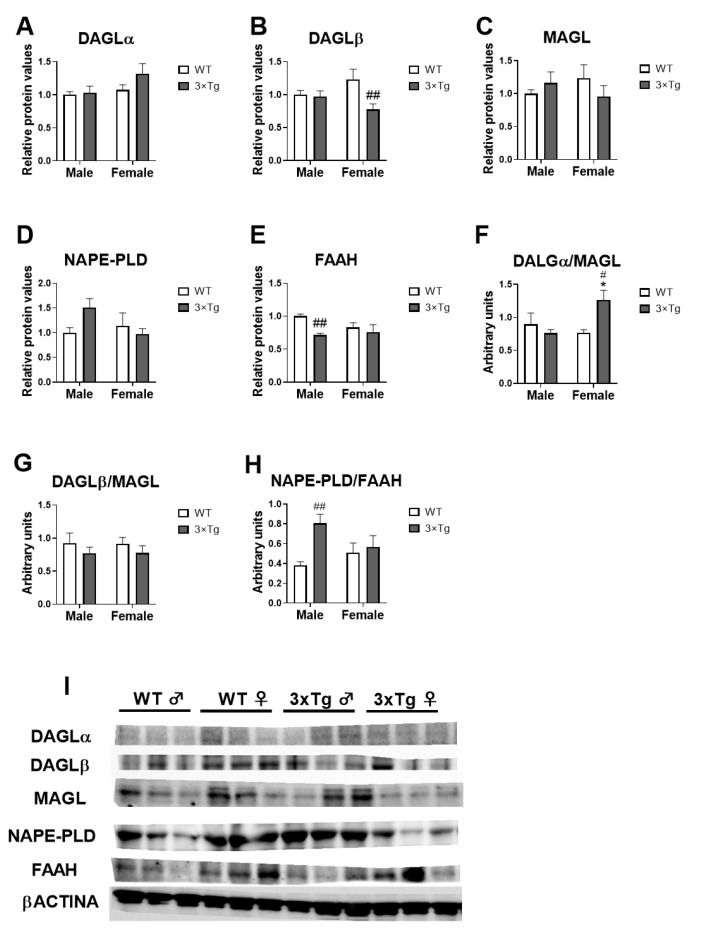
Cannabinoid enzymes protein expression: *Dagla* (**A**), *Daglb* (**B**), *Mgll* (**C**), *Nape-pld* (**D**), *Faah* (**E**), *Dagla/Mgll* (**F**), *Daglb/Mgll* (**G**), and *Nape-pld/Faah* (**H**) in hippocampal astrocytes from offspring born to 3×Tg-AD and non-Tg (WT) mice. Representative immunoblots (**I**). Data are expressed as the mean ± S.E.M. Tukey or single-effect analysis: * *p* < 0.05 vs. males (sex difference); ^#/##^
*p* < 0.05/0.01 vs. same-sex WT group.

**Figure 5 ijms-24-12598-f005:**
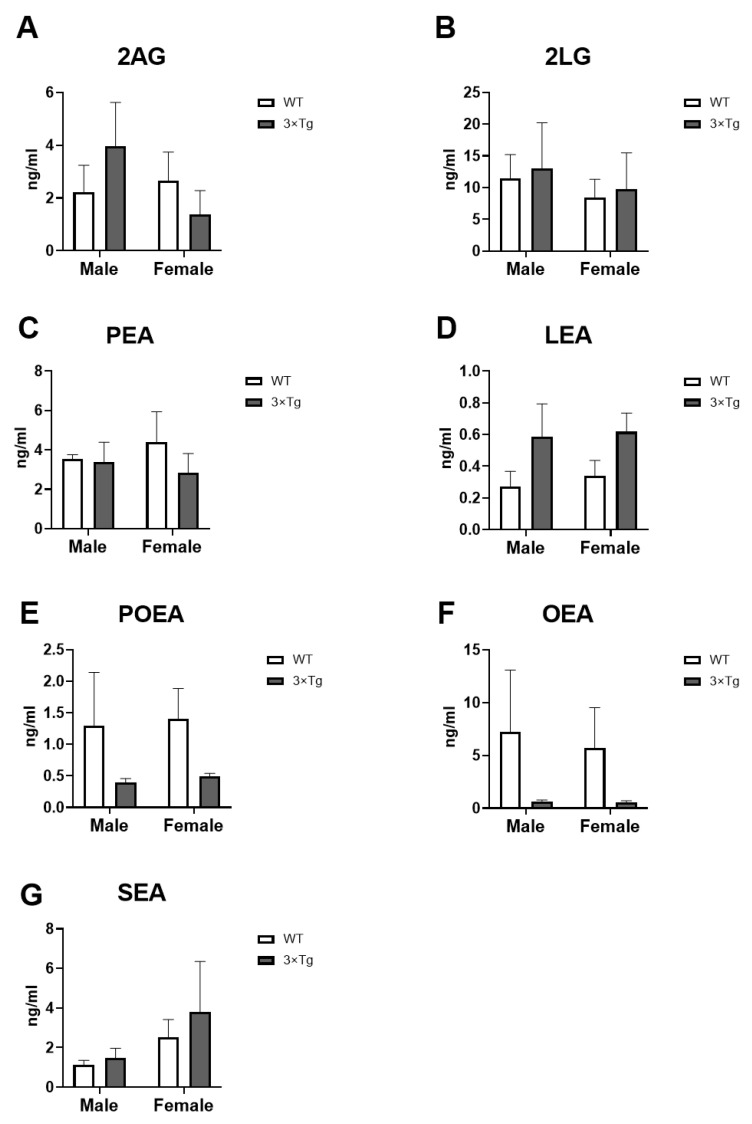
Endocannabinoid levels: 2AG (**A**), 2LG (**B**), POEA (**C**), LEA (**D**), PEA (**E**), OEA (**F**), and SEA (**G**) in hippocampal astrocytes cultured medium from offspring born to 3×Tg-AD and non-Tg (WT) mice. Data are expressed as the mean ± S.E.M.

**Figure 6 ijms-24-12598-f006:**
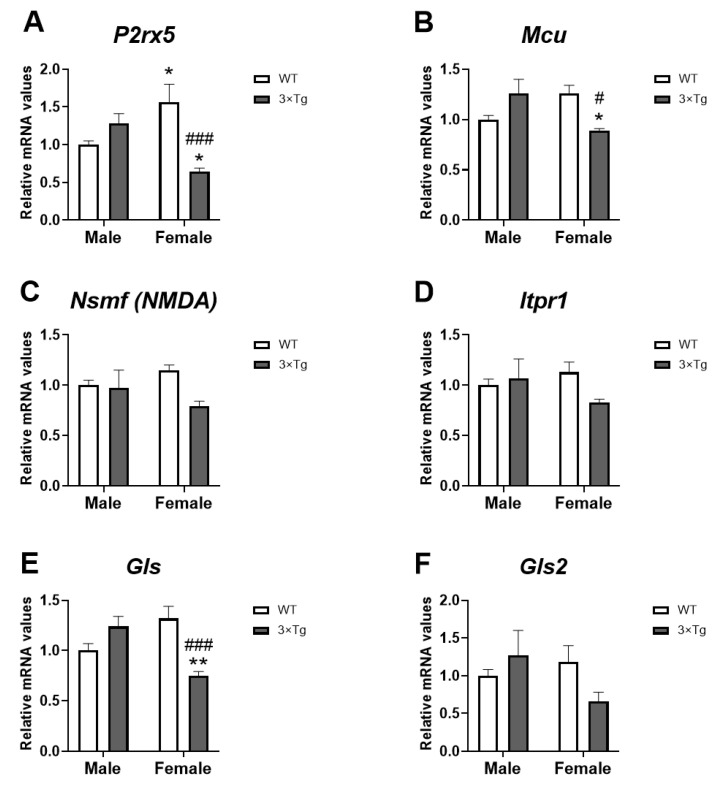
Ca^2+^ signaling mRNA expression: *P2rx5* (**A**), *Mcu* (**B**), *Nsmf* (**C**), *Itpr1* (**D**), *Gls* (**E**), and *Gls2* (**F**) in hippocampal astrocytes from offspring born to 3×Tg-AD and non-Tg (WT) mice. Data are expressed as the mean ± S.E.M. Tukey or single-effect analysis: */** *p* < 0.05/0.01 vs. males (sex difference); ^#/###^
*p* < 0.05/0.001 vs. same-sex WT group.

**Table 1 ijms-24-12598-t001:** Primer references for Taqman^®^ Gene Expression Assays.

Gene Symbol	Assay ID	GenBank Accession Number	Amplicon Length (bp)
*Actb*	Mm02619580	NM_007393.5	143
*Gfap*	Mm01253033	NM_001131020.1	75
*Vim (Vimentin)*	Mm01333430	NM_011701.4	62
*Tnfa*	Mm00443258	NM_001278601.1	81
*Il1b*	Mm00434228	NM_008361.3	90
*Gls*	Mm01257297	NM_001081081.2	114
*Gls2*	Mm01164862	NM_001033264.3	118
*Cnr1 (CB1)*	Mm01212171	NM_007726.3	66
*Cnr2 (CB2)*	Mm02620087	NM_009924.4	171
*Gpr55*	Mm02621622	NM_001033290.2	102
*Ppara*	Mm00440939	NM_001113418.1	74
*Trpv1*	Mm01246300	NM_001001445.2	56
*Trpa1*	Mm01227437	NM_177781.4	61
*Dagla*	Mm00813830	NM_198114.2	69
*Daglb*	Mm00523381	NM_144915.3	72
*Nape-pld*	Mm00724596	NM_178728.5	85
*Mgll*	Mm00449274	NM_001166249.1	78
*Faah*	Mm00515684	NM_010173.4	62
*P2rx5*	Mm00473677	NM_033321.3	104
*Mcu*	Mm01168773	NM_001033259.4	71
*Nsmf*	Mm00480341	NM_001039386.1	70
*Itpr1*	Mm00439907	NM_010585.5	58
*Il6*	Mm00446190	NM_031168.1	78
*Ptgs2*	Mm00478374	NM_011198.3	80

**Table 2 ijms-24-12598-t002:** Antibodies used for protein expression by Western blotting.

Antigen	Manufacturing	Dilution
βactin	Sigma (St. Louis, MO, USA) #2535L, Mouse monoclonal antibody.	1:2000
DAGLa	bioNova Científica (Madrid, España) (#orb156533). Rabbit polyclonal antibody	1:100
DAGLb	Biorbyt (Cambridge, United Kingdom)(orb182976). Rabbit polyclonal antibody	1:200
NAPE-PLD	Abcam (Cambridge, United Kingdom). Rabbit polyclonal antibody. Ab95397	1:200
MAGL	Abcam (Cambridge, United Kingdom). Rabbit polyclonal antibody. (Ab24701)	1:200
FAAH	Cayman (Ann Arbor, MI, USA). Rabbit polyclonal antibody. Cat. No: 101600	1:200

## Data Availability

The data that support the findings of this study are available from the corresponding author upon reasonable request.

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
