# Peer review of "Sex-Dependent Altered Expression of Cannabinoid Signaling in Hippocampal Astrocytes of the Triple Transgenic Mouse Model of Alzheimer’s Disease: Implications for Controlling Astroglial Activity"

_ijms, 2023, doi:10.3390/ijms241612598_

Round 1

Reviewer 1 Report

I thank the authors for their work in preparing this manuscript. Herein, the authors present gene expression data of crucial components of astrocyte activity, the endocannabinoid system, and calcium signaling. Additionally, mass spectrometric measurements of endocannabinoid 2-arachidonoyl glycerol and endocannabinoid-related molecules are provided. Using the 3x-Tg model of Alzheimer’s disease, the authors present data supporting the claim that sex-differences present at birth (post-natal day 2-3) may underly the development of AD. Unfortunately, throughout this work, gene expression data are described as an output for overall function and signaling. Without evaluation of post-translational modification, protein translation, phosphorylation status and other modifiers of receptor and, especially, enzyme function, it cannot be determined if gene expression status alone is sufficient to infer signaling alterations; specifically, it has been reported by several groups that gene expression of hydrolytic enzymes in the endocannabinoid system may not have an effect or even contradict functional outcomes. While useful for subcellular localization, the expression changes observed are not convincing that overall signaling is substantially altered in these mice, especially given the lack of other behavioral or phenotypic outcomes related to AD or calcium flux. The young age of these mice precludes many behavioral assessments, but also brings about a crucial concern regarding the study design. At this time, age is the primary risk factor in development of AD, however post-natal day 2-3 is far too early to detect many AD-related outcomes. These findings would be more impactful with addition of other genetic models of AD to confirm this result. As well, several language errors are detected throughout, including the misspelling of Signaling, shown as “Signalling” even in the title. I highly recommend the authors utilize grammar and spelling software to avoid sentence level inaccuracies that may hinder the ability of readers to understand the material. Overall, I applaud the authors for their efforts at disentangling the vital sex-dimorphism in AD and for beginning to tease apart the astrocyte-endocannabinoid system in AD development. Please see detailed concerns and sentence level suggestions below.

Line 74, description of other endocannabinoid hydrolyzing enzymes (e.g., ABHD6, ABHD12, and COX-2) are notably missing. Considering prostaglandin formation and downstream effects on microglia in transgenic models of Alzheimer’s disease (i.e., Johansson et al., JCI 2015, PMID 25485684), whether beneficial or deleterious, is crucial for given study.

Line 79, is the nature of astroglial modulation beneficial or detrimental. This requires elaboration to determine rationale for evaluating the ECS in astrocytes. As well, this paragraph comprises a single run-on sentence. Please separate appropriately.

Line 111, the authors state that the transgenic model is born with deficits in hippocampal astrocytes, however the relevance of disease state at birth in these mice and how it relates to AD, for which the greatest risk factor is age, is not clearly described. Do these mice show other pathologic features of AD at birth?

Results sections includes ANOVA results, however details of post hoc tests used for each analysis are missing. This is troubling, because figure 2B indicates a significant difference between male and female WT mice, however the error bars as displayed overlap. This suggests that the difference between these datasets may not be statistically significant. Quality of bar graphs all around would be improved by addition of individual datapoints. Figure legends indicate Tukey and single-effect post hoc tests, however which datasets were handled by which test is unclear.

Line 246, “…from 3xTg-AD mice at 2-3 postnatal days…”. This is the first mention of the age of the mice used in the study. This is a vital consideration that needs to be addressed in the Introduction and must be contextualized with each result.

Line 256, “Interestingly, these changes are mostly sex-dependent…”. More literature needs to be provided to describe apparent sex differences at postnatal days 2-3, timepoint of collection. Many sexually dimorphic features manifest during late childhood into adolescence in mammals. This has potential to be transformative for roles of sex chromosomes alone in AD development but suffers from a paucity of relevant background.

Line 276, this paragraph suggests that the astrocyte expression of CB1 in murine models of AD is controversial, however only a single specific direction is given and it does not synthesize finds from the current work. Please elaborate on the role for these findings given the changes observed.

Line 287, WIN55 212-2 is a dual CB1 and CB2 receptor agonists. Please re-evaluate the statements in this sentence.

Line 293, not all N-acylethanolamines are converted into arachidonic acid, including many of the assayed species (i.e., palmitoylethanolamide, oleoylethanolamide, etc.). Please revise this statement.

Line 365, the method for astrocyte separation requires more explanation. What method was used to confirm purity of primary cell culture and has this method been peer-reviewed?

Line 381, beta-actin (Actb) is used for normalization because there was no significant difference observed between groups. Please provide the relative expression of Actb. Although not significant, not enough information is provided to determine the suitability of Actb in this context. Significance is not a proxy for variability between datasets.

Line 388, HPLC-MS (high performance liquid chromatography tandem mass spectrometry) is referenced for the first time in text and the abbreviation must be explained.

Sentence Level Changes:

Line 48, separate “… ofbeta-amyloid…” into separate words.

Line 59, signaling is misspelled as “signalling”. Double check throughout. Also observed in line 65.

Line 78, add a comma after “i.e.” and remove extra period at the end of the sentence.

Line 92, change “improve” to “improved”.

Line 225, please reword for grammatical accuracy, “… being significant the lower gene expression of CB1 and CB2 receptors.”

Line 232, “… in in …” please revise.

Reviewer 2 Report

This is a well-written report on the status of the cannabinoid system in the most used animal model of AD, the 3xTg-AD mice. Information is relevant to the field since few reports of signaling of these receptors are currently available. On the other hand, moderately increasing but significant differences between sex are described that can explain some sex differences in AD incidence. It can be improved manuscript adding a draw about differences observed in transgenic and normal astrocytes. 

Reviewer 3 Report

In the current study titled "Sex-dependent altered expression of cannabinoid signaling in hippocampal astrocytes of triple transgenic mouse model of Alzheimer's disease: implication of controlling astroglial activity," the authors investigate cannabinoid signaling in hippocampal astrocytes from the 3xTg-AD mouse model.

The authors assess the expression of inflammatory markers and glial activity-related factors, including GFAP, Vimentin, Tnfα, Il1b, Il6, and ptgs (Cox2). They observe a significant difference in Il6 expression between wild-type and Tg animals, specifically among females. Additionally, the authors analyze the expression of cannabinoid-related receptors (Cnr1, Cnr2, Gpr55, Ppara, Trpv1, and Trpa1) and find downregulation of Cnr1 in Tg females. They also measure the mRNA expression of cannabinoid enzymes (Dagla, Daglb, MgII, Nap-pld, and Faah) and identify a significant difference in Daglb expression between wild-type and Tg females. Furthermore, they observe significant differences in Daglb/Mgll and Nap-pld/Faah ratios among Tg females compared to wild-type.

The authors measure the levels of endocannabinoids (2-Arachidonoylglycerol (2AG), 2-Linoleoylglycerol (2LG), palmitoleoylethanolamide (POEA), linoleylethanolamide (LEA), palmitoylethanolamide (PEA), oleoylethanolamide (OEA), and N-stearoylethanolamine (SEA)) in the culture medium.

Additionally, the expression levels of calcium signaling-related genes (P2rx5, Mcu, Nsmf (NMDA), Itpr1, Gls, and Gls2) are measured in the study.

Overall, the study is well-designed, systematic, and suitable for publication in the journal, with some minor issues that need to be addressed.

  1. The text should include the notation "ns" (not significant) in the graph to clearly indicate non-significant differences for easier reader understanding.
  2. Please verify the significance of the difference in expression of Trpa1 in Figure 2F and provide the corresponding p-value.
  3. In Figure 4A, 4C, 4D, and 4F, where there are large differences in protein levels, please provide the actual p-values to accurately convey the significance of these differences.
  4. Line 219 mentions a change in gene expression in the 3xTg-AD mouse, while the results indicate differences only among females. Please clarify and revise this statement accordingly.
  5. When discussing results in the discussion section, please reference the corresponding figure numbers for each result discussed to enhance clarity and facilitate cross-referencing.
  6. Please correct the word "ofbeta" in line 48.
